# Scene Graph Disentanglement and Composition for Generalizable Complex Image Generation

**Yunnan Wang[1,2]  Ziqiang Li[1,2]  Wenyao Zhang[1,2]**
**Zequn Zhang[2,3]  Baao Xie[2]  Xihui Liu[4]  Wenjun Zeng[2]  Xin Jin[2]***
[1]Shanghai Jiao Tong University, Shanghai, China
[2] Ningbo Institute of Digital Twin, Eastern Institute of Technology, Ningbo, China
[3]University of Science and Technology of China, Hefei, China
[4]The University of Hong Kong, Hong Kong, China
{wangyunnan, ziqiangli}@sjtu.edu.cn   jinxin@eitech.edu.cn

## Abstract

There has been exciting progress in generating images from natural language or layout conditions. However, these methods struggle to faithfully reproduce complex scenes due to the insufficient modeling of multiple objects and their relationships. To address this issue, we leverage the scene graph, a powerful structured representation, for complex image generation. Different from the previous works that directly use scene graphs for generation, we employ the generative capabilities of variational autoencoders and diffusion models in a generalizable manner, compositing diverse disentangled visual clues from scene graphs. Specifically, we first propose a Semantics-Layout Variational AutoEncoder (SL-VAE) to jointly derive *(layouts, semantics)* from the input scene graph, which allows a more diverse and reasonable generation in a one-to-many mapping. We then develop a Compositional Masked Attention (CMA) integrated with a diffusion model, incorporating *(layouts, semantics)* with fine-grained attributes as generation guidance. To further achieve graph manipulation while keeping the visual content consistent, we introduce a Multi-Layered Sampler (MLS) for an "isolated" image editing effect. Extensive experiments demonstrate that our method outperforms recent competitors based on text, layout, or scene graph, in terms of generation rationality and controllability. Code is available at https://github.com/wangyunnan/DisCo.

## 1 Introduction

Text-to-image (T2I) generation with diffusion models (DMs) [1, 2] has yielded remarkable advancements [3, 4, 5] in recent years, benefiting from the developments of vision-language foundation models [6, 7, 8, 9]. However, textual conditions with linear structure struggle to delineate the intricacies of complex scenes precisely. For example, as shown in the failure cases of DALL·E 3 [3] in Figure 1 (a), given the intricate text prompt "*A sheep by another sheep ... a boat on the grass.*", the T2I model may have difficulty accurately generating object relationships or quantities. Consequently, some studies [10, 11, 12, 13] strive to improve spatial relationship (e.g., "*by*" and "*on*") control by incorporating additional layout conditions. Nevertheless, as illustrated in the failure cases of LayoutDiffusion [10] in Figure 1 (b), layout-to-image (L2I) methods inevitably encounter challenges in representing certain non-spatial interactions, such as depicting "*playing*" within spatial topology.

To efficiently depict complex scenes for guiding generative models, recent methods [14, 15, 16] utilize structured scene graphs as conditions instead of text or layout prompts. Scene graphs [17]

---

*Xin Jin is the corresponding author.

38th Conference on Neural Information Processing Systems (NeurIPS 2024).

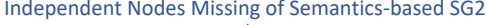

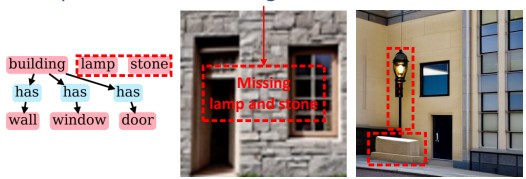

(a) DALL·E 3 (middle) and Ours (right).

(b) LayoutDiffusion (middle) and Ours (right).

(c) R3CD (middle) and Ours (right).

(d) Results w/o (middle) and w/ (right) AC.

Figure 1: Failure cases generated by **(a)** text-to-image (T2I) (DALL·E 3 [3]), **(b)** layout-to-image (L2I) (LayoutDiffusion [10]), and **(c)** semantics-based scene-graph-to-image (SG2I) (R3CD [14]) methods . **(d)** Generalizable object *Attribute Control* (AC) under consistency achieved by our DisCo.

represent scenes with a structured graph format, where objects within the scene are denoted as nodes and the relationships between objects are represented as edges. Scene-Graph-to-Image (SG2I) generation is a challenging task due to the frequent ambiguous alignment between graph edges and relationships/interactions among visual objects. To address this issue, layout-based SG2I methods [15, 17, 18, 19] explicitly predict the spatial arrangements of objects in scenes by additional layout predictors, followed by L2I synthesis according to the layout (as demonstrated in Figure 2 (a)). These methods commonly employ one-to-one mapping, i.e., a single scene graph only corresponds to one layout, which severely limits the generation diversity. Besides, they also inherit the limitation of the L2I approach in modeling non-spatial interactions, whereby each object is typically generated independently. In contrast, as shown in Figure 2 (b), semantic-based SG2I methods implicitly encode graph edges into node embeddings by graph convolutional networks (GCNs), which effectively aligns object semantics with non-spatial interactions. Nonetheless, these methods are weak in logically determining the spatial positions of independent nodes, which might cause the absence of independent nodes (e.g., the "*lamp*" and "*stone*" shown in Figure 1 (c)) in the generated image.

In this paper, we propose **DisCo**, a **Co**mpositional image generation framework that integrates the **Dis**entangled layout and semantics derived from scene graph representations (as depicted in Figure 2 (c)). To boost the representational capacity of scene graphs for complex scenes, we augment the node and edge representations with CLIP [6] text embeddings, and incorporate extra spatial information (i.e., bounding box embeddings) for nodes during training. Once the textual scene graph is constructed, we propose a *Semantics-Layout Variational AutoEncoder* (SL-VAE) based on triplet-GCN [18] to jointly model the spatial relationships and non-spatial interactions in the scene. SL-VAE allows the one-to-many disentanglement for *spatial layout* and *interactive semantics* that match the

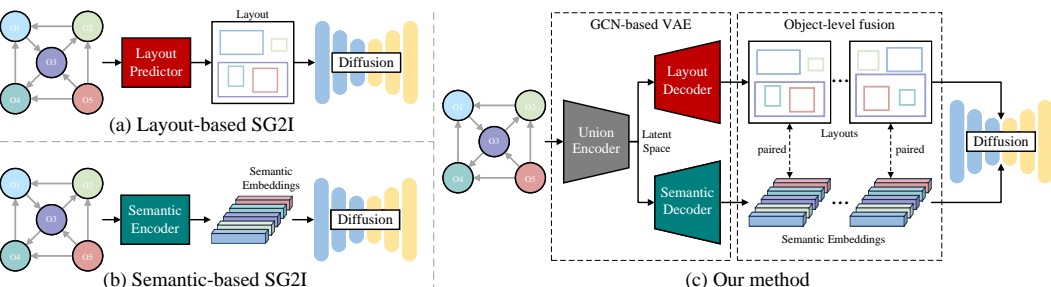

Figure 2: **Comparison between the previous SG2I architectures and ours**. **(a)** Layout-based SG2I model [15] generate a spatial arrangement with an object layout; **(b)** Semantic-based SG2I models [14, 16] build interactive semantic embedding between objects; **(c)** Our method leverages scene graph representation by jointly deriving the disentangled layout and semantics with the proposed SL-VAE.

input scene graph. This is achieved by sampling from a Gaussian distribution, offering object-level *(layouts, semantics)* conditions for the diffusion process [20]. Given that the layout and semantics encapsulate global and local relational information, we further introduce the *Compositional Masked Attention* (CMA) mechanism to inject object-level graph information with fine-grained attributes into the diffusion model, thereby preventing relational confusion and attribute leakage.Finally, we present a *Multi-Layered Sampler* (MLS) technique that leverages the diverse conditions generated by SL-VAE, achieving generalizable generation for object-level graph manipulation (i.e., node addition and attribute control) in the SG2I task, as depicted by the color change of two "*sheep*" in Figure 1 (d).

In summary, our key contributions are as follows: **(i)** We apply the textual scene graph as a structured scene representation and introduce the *Semantics-Layout Variational AutoEncoder* (SL-VAE) to disentangle diverse *spatial layouts* and *interactive semantics* from the scene graph; **(ii)** We present the *Compositional Masked Attention* (CMA) to inject extracted object-level graph information with fine-grained attributes into the diffusion model, which avoids relational confusion and attribute leakage; **(iii)** We introduce the *Multi-Layered Sampler* (MLS), a technique that leverages the diverse conditions produced by SL-VAE to implement object-level graph manipulation while keeping the visual content consistent; **(iv)** Our method outperforms current text/layout-based methods in relationship generation and achieves significantly superior generation performance compared to state-of-the-art SG2I models, thus showcasing the generalization of textual scene graphs in depicting complex scenes.

## 2 Preliminary

### 2.1 Text-to-Image Diffusion Models

Diffusion models (DMs) [1] are generative models that learn the data distribution $p(\boldsymbol{x})$ by gradually performing $T$-step noise reduction from the variables $\boldsymbol{x}_T$ sampled from the Gaussian distribution $\mathcal{N}(0,1)$. Thus the training process of DMs can be regarded as the reverse process of a Markov chain with a fixed length $T$. To generate high-resolution images with less computational resources, Latent Diffusion Models (LDMs) [20] encode the image $\boldsymbol{x}$ into the latent space $\boldsymbol{z}$ with the pre-trained Vector Quantized Variational AutoEncoder (VQ-VAE). Subsequently, the LDMs aim to predict the distribution $p(\boldsymbol{z})$ rather than $p(\boldsymbol{x})$. For the text-to-image LDMs, the text is encoded with the CLIP [6] text encoder $E_{\mathrm{CLIP}}$. Then the objective function of a text-guided LDM can be formulated as follows:

$$\mathcal{L}_{LDM} = \mathbb{E}_{\boldsymbol{z},\boldsymbol{\epsilon}\sim\mathcal{N}(0,1),t}[\|\boldsymbol{\epsilon} - \boldsymbol{\epsilon}_\theta(\boldsymbol{z}_t, E_{\mathrm{CLIP}}(\boldsymbol{c}),t)\|_2^2], \tag{1}$$

where $E_{\mathrm{CLIP}}(\boldsymbol{c})$ is the text embedding of the text condition $\boldsymbol{c}$, $t$ is the diffusion step, $\boldsymbol{\epsilon}_\theta$ is a model for estimating the noise $\boldsymbol{\epsilon}$, and $\boldsymbol{z}_t = \alpha_t \boldsymbol{z}_{t-1} + \sigma_t \boldsymbol{\epsilon}$ is the $t$-step noised latent code from the ground-truth $\boldsymbol{z}_0$. During inference, the model $\boldsymbol{\epsilon}_\theta$ with various samplers [1, 21] gradually denoises the initial noise $\boldsymbol{z}_T \sim \mathcal{N}(0,1)$. Finally, the predicted latent code is decoded into the image space.

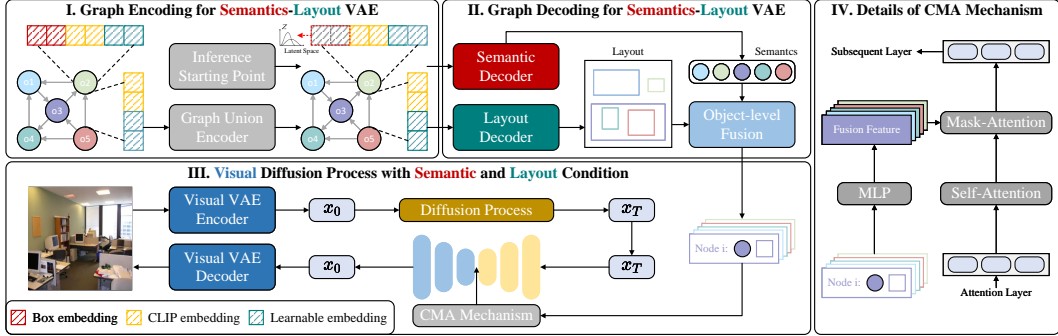

Figure 3: **Framework overview**. (I) We parameterize the node embeddings into the Gaussian distribution with the Graph Union Encoder, which jointly models the spatial relationships and non-spatial interactions in scene graphs; (II) The Semantic and Layout Decoders generate spatial layouts and interactive semantics sampled from Gaussian distribution, respectively; (III) A diffusion model with the proposed Compositional Masked Attention (CMA) incorporates object-level conditions to generate visual images following the scene graph description; (IV) Detailed structure of CMA Layer.

## 2.2 Scene Graph Representation

The scene graph $G = (O, E)$ [17] presents a structured scene representation. Nodes $O = \{o_i\}_{i=1}^{N_o}$ denotes $N_o$ objects within the scene, while edges $E = \{e_{ij}\}_{1 \leq i,j \leq N_o, i \neq j}$ denotes relationships between objects. All nodes and edges come with a semantic label, denoted as $c_i^o \in \mathcal{C}^o$ and $c_{ij}^e \in \mathcal{C}^e$, where $\mathcal{C}^o$ and $\mathcal{C}^e$ are category vocabularies of nodes and edges, respectively. In practice, the nodes $O = \{o_i\}_{i=1}^{N_o}$ and the triples $T = \{t_{ij} = (o_i, e_{ij}, o_j)\}_{1 \leq i,j \leq N_o, i \neq j}$ representing connections from $o_i$ to $o_j$ serve as inputs for graph convolutional networks (GCNs). Moreover, nodes and edges are typically converted into learnable embeddings using embedding layers denoted as $E_{emb}^o$ and $E_{emb}^e$.

## 3 Methodology

As illustrated in Figure 3, we present a novel SG2I synthesis framework known as DisCo. The DisCo comprises three primary components: (1) Semantics-Layout Variational AutoEncoder (SL-VAE) that disentangle diverse spatial layouts and interactive semantics from the scene graph (Section 3.1); (2) Compositional Masked Attention (CMA) that injects object-level *(layouts, semantics)* with fine-grained attributes into the diffusion model (Section 3.2); and (3) Multi-Layered Sampler (MLS) that implements generalizable generation for object-level graph manipulation (Section 3.3).

### 3.1 Semantics-Layout Variational AutoEncoder

**Textual Scene Graph Construction**. For constructing the scene graph representation, we employ the visual-language model to fully leverage the inherent disentangled semantics of the language, while simultaneously facilitating the alignment between images and scene graphs. Specifically, we augment the node and edge embeddings of the scene graph with CLIP [6] text embeddings. During training, we also incorporate spatial information for node embeddings by including bounding box coordinates (i.e., top-left corner and box size denoted as $b_i = (x_i, y_i, w_i, h_i)$). Then the node embeddings $\mathcal{O}$ and edge embeddings $\mathcal{E}$ can be formulated as follows:

$$\mathcal{O} = \{E_{emb}^o(c_i^o) \otimes E_{\mathrm{CLIP}}(o_i) \otimes E_{box}(b_i)\}_{i=1}^{N_o}, \ \mathcal{E} = \{E_{emb}^e(c_{ij}^e) \otimes E_{\mathrm{CLIP}}(t_{ij})\}_{1 \leq i,j \leq N_o, i \neq j} \quad (2)$$

where $E_{\mathrm{CLIP}}$ denotes the frozen pre-trained text encoder, $E_{box}$ is the spatial encoder for bounding box coordinates using Multi-Layer Perceptions (MLPs), and $\otimes$ denotes concatenate operation.

**Graph Union Encoding**. Although layout-based SG2I methods are superior in modeling spatial topology compared to the semantics-based method, they fall short in capturing object interactions (i.e., non-spatial relationships) within the scene. Accordingly, after obtaining the node and edge embeddings mentioned above, we apply a Conditional Variational Autoencoder (CVAE) [22] based on triplet-GCN [18] to jointly model the layout and semantics information. As shown in Figure 3.I, the $L$-layer Graph Union Encoder $E_u$ takes node and edge embeddings as inputs:

$$(\phi_i^{l+1}, \phi_{ij}^{l+1}, \phi_j^{l+1}) = \mathrm{GCN}_l(\phi_i^l, \phi_{ij}^l, \phi_j^l), l \in \{0, \dots, L-1\} \quad (3)$$

where $l$ denotes the layer index of Graph Union Encoder, and $\phi$ denotes intermediate features. Here we initialize $(\phi_i^0, \phi_{ij}^0, \phi_j^0) = (\mathcal{O}_i, \mathcal{E}_{ij}, \mathcal{O}_j)$. Please refer to the **Appendix** for more details about the triplet-GCN. Given that the last node embedding $\phi_i^L$ integrates both topology and interaction information, we conduct layout-semantic modeling by parameterizing it into Gaussian spaces $Z \sim \mathcal{N}(\mu, \sigma)$. In this context, the means $\mu \in \mathbb{R}^{D_z}$ and variances $\sigma \in \mathbb{R}^{D_z}$ are estimated individually by two supplementary MLPs, where $D_z$ denotes the dimensional of latent space for node embedding. Hence, we jointly model the layout and semantics through the following minimization:

$$\mathcal{L}_{union} = \mathrm{KL}\left(E_u(u|y, \mathcal{O}, \mathcal{E}) \parallel p(u|y)\right), \quad (4)$$

where KL denotes the Kullback-Liebler divergence, $y$ denotes condition and the prior $p(u|y)$ is the standard Gaussian distribution $\mathcal{N}(u \mid 0, 1)$. Specifically, we condition the latent space of the graph structure using the edge embedding following Equation 2 alongside the updated node embedding $\{E_{emb}^o(c_i^o) \otimes E_{\mathrm{CLIP}}(o_i) \otimes u_i\}_{i=1}^{N_o}$, where $u_i$ is a random vector sampled from $Z$. This architecture ensures that layout is solely necessary for training, with no need for hand-crafted layout in inference.

**Disentangled Semantics-Layout Decoding**. As illustrated in Figure 3.II, we disentangle the explicit *spatial layout* and implicit *interactive semantics* from the latent space using two separate triplet-GCN-based decoders, i.e., layout decoder $D_l$ and semantic decoder $D_s$. The proposed Semantics-Layout

Variational AutoEncode (SL-VAE) comprises these two decoders and the graph union encoder mentioned above, which derives the spatial topology and object interactions from the scene graph representation. During training, the layout decoder is optimized by the following objective function:

$$\mathcal{L}_{layout} = \frac{1}{N_o} \sum_{i=1}^{N_o} |b_i - \hat{b}_i|_1, \tag{5}$$

where $\hat{b}_i$ denotes the predicted coordinates. We only incorporate the ground truth layout $\mathcal{B} = \{b_i\}_{i=1}^{N_o}$ during training, while generating $N_l$ diverse *layouts* $\{\hat{\mathcal{B}}_n = \{\hat{b}_{n,i}\}_{i=1}^{N_o}\}_{n=1}^{N_l}$ by sampling Gaussian noise at inference time. For simplicity, we omit the superscript ˆ in the following description. The semantic decoder $D_s$ generates *semantics* embeddings $\mathcal{S} = \{s_i\}_{i=1}^{N_o}$ to facilitate subsequent diffusion processes, and its parameters are iteratively updated with the diffusion loss in the next Section 3.2.

### 3.2 Diffusion with Compositional Masked Attention

**Object-level Fusion Tokenizer**. We integrate the *spatial layout* $\mathcal{B} = \{b_i\}_{i=1}^{N_o}$ and *interactive semantics* $\mathcal{S} = \{s_i\}_{i=1}^{N_o}$ at object level, as illustrated in Figure 3.III. The single-object embeddings $\mathcal{C} = \{c_i\}_{i=1}^{N_o} = \{s_i \otimes \mathcal{F}(b_i)\}_{i=1}^{N_o}$ are acquired by directly applying semantic embeddings, while encoding box information using a Fourier mapping $\mathcal{F}$ [23]. We define a learnable null embedding to pad the embedding length to $N_{max}$, thereby accommodating varying numbers of objects:

$$c_i = \begin{cases} s_i \otimes \mathcal{F}(b_i), & i \leq N_o \\ c_{null}, & \text{otherwise} \end{cases} \tag{6}$$

where $c_{null}$ denotes the learnable null embedding for padding. We optionally add attribute embedding $\mathcal{A} = \{a_i\}_{i=1}^{N_{max}}$ to construct updated $\mathcal{C} = \{c_i \otimes a_i\}_{i=1}^{N_{max}}$, where $c_i$ and $a_i$ are separately processed by two MLPs before concatenation. Note that $a_i$ is obtained similarly to edge embedding in Equation 2. We also define a learnable null embedding $a_{null}$ for cases where no attribute is specified.

**Compositional Masked Attention**. The cross-attention mechanism in diffusion bridges the visual and textual information, while self-attention captures self-related information within visual tokens [24]. Therefore, we insert our proposed Compositional Masked Attention (CMA) between self-attention and cross-attention layers. This technique effectively injects graph information into the diffusion process at the object level, preventing semantic confusion and attribute leakage through the attention mask. Specifically, we denote the visual token output by the vanilla self-attention as $\mathcal{V} \in \mathbb{R}^{N_v \times D_v}$, where $N_v$ and $D_v$ represent the number and dimensions of tokens, respectively. Then the CMA layer can be expressed as:

$$\hat{\mathcal{V}} = SA_{mask}(\mathcal{V} \otimes \hat{\mathcal{C}}, \mathbf{M})[: N_v], \tag{7}$$

where $\hat{\mathcal{C}} = \{\hat{c}_i\}_{i=1}^{N_{max}}$ denotes object embeddings whose dimensions are aligned with the visual token $\mathcal{V}$ using MLPs. The matrix $\mathbf{M} \in \mathbb{R}^{(N_v + N_{max}) \times (N_v + N_{max})}$ denotes the attention mask that depends on layout $\mathcal{B}$, which can be constructed as follows:

$$\mathbf{M}_{i,j} = \begin{cases} 1, & \text{if } i, j \text{ fall into the same object} \\ -inf, & \text{otherwise} \end{cases} \tag{8}$$

where "$i, j$ fall into the same object" means that $i$ and $j$ index the visual tokens or object embeddings of the same object. Figure 4 illustrates the mechanism of CMA through a toy example. In contrast to vanilla self-attention, the proposed CMA prevents relational confusion and attribute leakage between different objects through the well-designed object-level masks mentioned above. As shown in Figure 3.IV, we forward the output of the CMA layer into the subsequent layers, serving as the updated visual token.

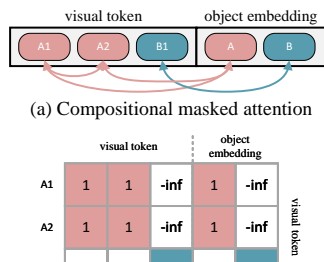

(a) Compositional masked attention

(b) Attention mask

Figure 4: **Toy example** of (a) compositional masked attention, and (b) its corresponding attention mask. We use visual tokens and object embeddings of objects A and B for demonstration. A and B have 1 and 2 visual tokens, respectively, whose attribution is determined by bounding boxes.

**Diffusion Loss**. Based on the object semantics output from the proposed SL-VAE, we optimize the evidence lower bound between sampling noise and prediction noise conditioned on object-level

Table 1: **Performance comparison** on COCO-Stuff and Visual Genome datasets using Inception Score (IS) and Fréchet Inception Distance (FID) metrics. We report the results of methods with two generator structures, namely GAN- and Diffusion-based. The architecture of these methods is based on the layout (L) or semantics (S), while our approach includes both. The best results are **bolded**.

| Method | Type | COCO [26] | | Visual Genome [27] | |
|---|---|---|---|---|---|
| | | IS ↑ | FID ↓ | IS ↑ | FID ↓ |
| Real Image | - | 30.7 | - | 27.3 | - |
| **GAN-based** | | | | | |
| SG2Im [17] | L | 8.2 | 99.1 | 7.9 | 90.5 |
| PasteGAN [28] | L | 12.3 | 79.1 | 8.1 | 66.5 |
| SOAP [18] | L | 14.5 | 81.0 | - | - |
| WSGC [19] | L | 6.5 | 121.7 | 9.8 | 84.1 |
| KCGM [29] | S | - | - | 11.6 | 27.4 |
| **Diffusion-based** | | | | | |
| LDM [20] | S | 22.2 | 63.8 | 16.5 | 45.7 |
| SGDiff [16] | S | 17.8 | 36.2 | 16.4 | 26.0 |
| SceneGenie [15] | L | 22.2 | 63.3 | 20.3 | 42.2 |
| R3CD [14] | S | 19.5 | 32.9 | 18.9 | 23.4 |
| DisCo (ours) | L+S | **23.1** | **30.8** | **22.3** | **21.9** |

information (i.e., ground truth spatial layout and generated interactive semantics). Then the training loss of the diffusion model equipped with CMA can be summarized as follows:

$$\mathcal{L}_{LDM} = \mathbb{E}_{\boldsymbol{z}, \boldsymbol{\epsilon} \sim \mathcal{N}(0,1), t}[\|\boldsymbol{\epsilon} - \boldsymbol{\epsilon}_\theta(\boldsymbol{z}_t, E_{\text{CLIP}}(O), \mathcal{B}, \mathcal{S}, \mathcal{A}, t)\|_2^2]. \quad (9)$$

Finally, we employ an end-to-end joint training pipeline for the whole proposed DisCo framework. The total objective function is presented as follows:

$$\mathcal{L}_{total} = \lambda_1 \mathcal{L}_{LDM} + \lambda_2 \mathcal{L}_{union} + \lambda_3 \mathcal{L}_{layout}, \quad (10)$$

where $\lambda_1$, $\lambda_2$, and $\lambda_3$ are hyperparameters, which are typically set to 1.0, 0.1 and 1.0, respectively.

### 3.3 Multi-Layered Sampler

Manipulations in the input scene graph, such as node addition and attribute adjustment, pose challenges for maintaining visual consistency in the generated images, ultimately compromising generalizability. To achieve an "isolated" image editing effect, we also provide the Multi-Layered Sampler (MLS) motivated by SceneDiffusion [25]. The scheme defines each object as a layer, thus allowing independent object-level Gaussian sampling. In contrast to SceneDiffusion which scrambles the reference layouts randomly, we sample additional $N_l$ (layouts, semantics) by the SL-VAE. Note that $N_l$ fixed seeds exist for the same scene. Then we aggregate latent codes from various layers into $\boldsymbol{z}_n$ and utilize layout-converted non-overlapping masks $\{\mathcal{M}_n = \{m_{n,i}\}_{i=1}^{N_o}\}_{n=1}^{N_l}$ for locally conditioned diffusion. During the inference, the noise estimation for $N_l$ scenes is calculated as follows:

$$\hat{\boldsymbol{\epsilon}}_n^{(t)} = \sum_{i=1}^{N_o} m_{n,i} \odot \boldsymbol{\epsilon}_\theta(\boldsymbol{z}_n^{(t)}, E_{\text{CLIP}}(o_i), b_{n,i}, s_{n,i}, a_i, t), \quad (11)$$

Subsequently, the latent code for each object is computed as the weighted average of the $N_l$ cropped denoised views. Please refer to the **Appendix** for more details about MLS.

## 4 Experiments

**Dataset**. We conduct scene-graph-to-image (SG2I) generation experiments on the Visual Genome (VG) [27] and COCO-Stuff (COCO) [26] datasets. The VG dataset comprises $108,077$ image-scene graph pairs, accompanied by the bounding box coordinates and object attributes. Following previous work [17], we select objects and relationships that appear at least $2,000$ and $500$ times respectively in VG, resulting in 178 objects and 45 unique relationship types. Also, we ignore small objects and use images containing 5 to 30 objects along with a minimum of 3 relationships. Based on the above

filtering, we have $62,565$ images available for training, each containing an average of 10 objects and 5 relationships. While the original COCO-Stuff dataset [26] lacks scene graph annotations, it consists of $40,000$ images annotated with bounding box coordinates and captions, essential for synthesizing geometric scene graphs [15, 16]. All images in the COCO-Stuff dataset are labeled as 80 item categories and 91 stuff categories.

**Implementation Details**. We fine-tune the pre-trained Stable-Diffusion 1.5[1] with the modified Attention module on 4 NVIDIA A100 GPUs, each with 80GB of memory. We apply the CLIP text encoder (vit-large-patch14 ) to construct the textual scene graph. We train the model with a batch size of 64 using the AdamW optimizer [30] with an initial learning rate of $1.0 \times 10^{-4}$, which is adjusted linearly over $50,000$ steps. During inference, we use the 50-step PNDMScheduler [21] with a classifiers-free scale [31] of 7.5. The sample number $N_l$ in the multi-layered sampler is set to 5.

**Evaluation Metrics**. Following previous works [14, 15, 16], we evaluate the performance of our method with the Inception Score (IS) [32] and the Fréchet Inception Distance (FID) [33]. The IS score is derived from a pre-trained Inception Net [34], assessing both the quality and diversity of synthesized images. The FID score quantifies the dissimilarity between the generated image and the real image distribution, which evaluates the fidelity of the generated images. To measure the effectiveness of compositional generation, we further evaluate our method on the T2I-CompBench [35]. Besides, we apply CLIP for zero-shot attribute classification of the controlled object cropped by the bounding box, and subsequently evaluate the attribute control performance by the classification accuracy $\text{ACC}_{attr}$.

**Quantitative Comparisons**. To demonstrate the effectiveness of the proposed DisCo, we compare it with current state-of-the-art SG2I methods on the COCO-Stuff and Visual Genome datasets, which are summarized in Table 1. Our DisCo outperforms other methods in both IS and FID scores, revealing its superior performance in both fidelity and diversity of image generation. Compared with previous methods, the primary architectural advantage of DisCo is

Table 2: **Relationship and attribute generation** compared with text-to-image methods on T2I-CompBench [35].

| Method | UniDet | CLIP | B-VQA | 3-in-1 |
|---|---|---|---|---|
| SD-v1.4 [20] | 0.1246 | 0.3079 | 0.3765 | 0.3080 |
| SD-v2 [20] | 0.1342 | 0.3127 | 0.5065 | 0.3386 |
| Composable [36] | 0.0800 | 0.2980 | 0.4063 | 0.2898 |
| Structured [37] | 0.1386 | 0.3111 | 0.4990 | 0.3355 |
| Attn-Exct [38] | 0.1455 | 0.3109 | 0.6400 | 0.3401 |
| GORS [35] | 0.1815 | 0.3193 | 0.6603 | 0.3328 |
| DisCo (ours) | **0.2376** | **0.3217** | **0.6959** | **0.4143** |

its innovative approach of simultaneously integrating disentangled layout and semantics extracted from scene graph representations. Moreover, the proposed SL-VAE achieves the diverse generation of layouts and semantics from a single scene graph through Gaussian distribution sampling. Therefore, our DisCo integrates the benefits of both layout-based and semantic-based methods, which is further ablated in detail in Table 4 of the ablation study. We proceed to assess the compositional generation on the T2I-CompBench [35], as shown in Table 2. The benchmark evaluates the competency of the text-to-image model in responding to compositional prompts. We report UniDet, CLIP, B-VQA, and 3-in-1 scores for measuring the generation of spatial/non-spatial relationships, attributes, and complex scenes, respectively. Following [35], we use the UniDet [39], CLIP [6], and BLIP [7] to evaluate these results. Our DisCo surpasses all compared T2I methods, confirming the efficacy of scene graphs in depicting complex scenes.

Table 3: **User study**. The score quantifies the user evaluation (i.e., relationships, quantities, and generation quality) of the alignment between the given prompt and the generated image.

| Method | SD-XL [40] | DALL·E 3 [3] | Imagen 2 [41] | GLIGEN [12] | LD [10] | MIGC [13] | SG2Im [17] | SGDiff [16] | R3CD [14] | DisCo (ours) |
|---|---|---|---|---|---|---|---|---|---|---|
| Score | 0.6684 | 0.5944 | 0.5637 | 0.6549 | 0.6200 | 0.7055 | 0.3783 | 0.4717 | 0.6928 | **0.8533** |

**User Study**. We conduct a user study by recruiting 50 participants from Amazon Mechanical Turk. We randomly select 8 prompts for each method, resulting in 80 generated images. We ask participants to score each generated image independently based on the image-prompt alignment. The worker can choose a score from $\{1, 2, 3, 4, 5\}$ and we normalize the scores by dividing them by 5. We then compute the average score across all images and all workers. The results are presented in the Table 3. Our method is favored by most participants in terms of generation rationality and controllability.

---

[1] https://huggingface.co/runwayml/stable-diffusion-v1-5

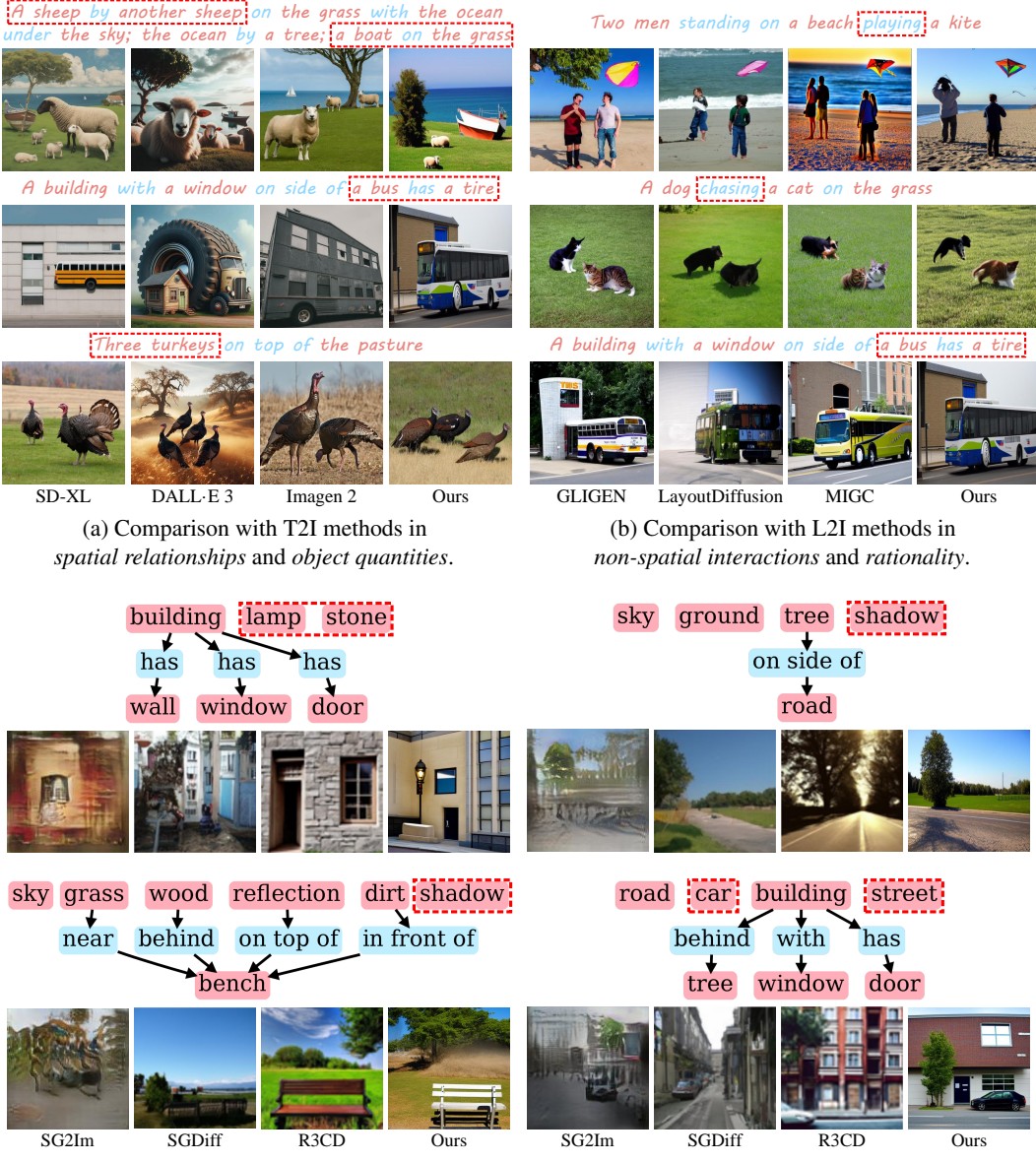

(a) Comparison with T2I methods in *spatial relationships* and *object quantities*.

(b) Comparison with L2I methods in *non-spatial interactions* and *rationality*.

(c) Comparison with SG2I methods in *independent node inference* and *generation quality*.

Figure 5: **Qualitative Comparisons** with **(a)** text-to-image (T2I) (SableDiffusion-XL [40], DALL·E 3 [3], and Imagen 2 [41]), **(b)** layout-to-image (L2I) (GLIGEN [12], LayoutDiffusion [10], and MIGC [13]), and **(c)** scene-graph-to-image (SG2I) (SG2Im [17], SGDiff [16], and R3CD [14]) methods.

Table 4: **Ablation study** for overall architecture. SL-VAE w/o $D_s$ means independent use of $\mathcal{O}$.

| Method | G2I-ACC ↑ | I2G-ACC ↑ |
|---|---|---|
| Layout ($D_l$) | 70.3 | 70.5 |
| Semantics ($D_s$) | 71.1 | 71.5 |
| SL-VAE (w/o $D_s$) | 72.9 | 72.8 |
| SL-VAE ($D_l + D_s$) | **73.9** | **74.3** |

Table 5: **Ablation study** for attention mechanism. Vanilla attention means off-the-shelf T2I attention.

| Attention Type | IS ↑ | FID ↓ |
|---|---|---|
| Vanilla attention | 17.2 | 29.1 |
| CMA (w/o mask **M**) | 17.9 | 28.4 |
| CMA (union MLP) | 19.8 | 22.0 |
| CMA (separate MLP) | **22.3** | **21.9** |

**Qualitative Comparisons**. Figure 5 visualizes the results of the methods conditioned by text, layout, or scene graph, showcasing our advantages in generating rationality and controllability: **(i)** *Comparison with the text-to-image (T2I) methods*. In Figure 5 (a), we present the superiority of

the disentangled structured scene graph over linear text for representing complex scenes. Firstly, we resolve ambiguity in textual relationships and semantics by employing layout and semantic disentanglement within the scene graph. For example, our DisCo clarifies the relationship between "boat" and "grass" in the first line, as well as the semantics of "bus" and "building" in the following

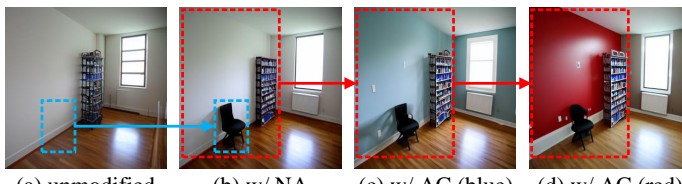

| (a) unmodified | (b) w/ NA | (c) w/ AC (blue) | (d) w/ AC (red) |

Figure 6: **Illustration** of object-level Node Addition (NA) and Attribute Control (AC) in the scene. From left to right: **(a)** the image generated by the unmodified scene graph; **(b)** the chair addition; **(c)** the blue-colored wall; and **(d)** the red-colored wall.

line. Additionally, the samples in the first and last lines also showcase our capacity to generate the specified quantities of objects precisely. (**ii**) *Comparison with the layout-to-image (L2I) methods*. We also demonstrate that our DisCo outperforms the diffusion methods relying on manually crafted scene layout representations, as illustrated in Figure 5 (b). While the L2I method struggles to model non-spatial interactions (such as "playing" in the first line and "chasing" in the second line), our DisCo addresses this challenge using disentangled object interactive semantics. Furthermore, by establishing semantics among objects derived from their relationships, we prevent independent generation instances that rely solely on the layout, exemplified by the "bus" and "tire" in the last line. (**iii**) *Comparison with the scene-graph-to-image (SG2I) methods*. The SG2I visualization results of different methods are showcased in Figure 5 (c). Our DisCo significantly improves the quality of SG2I generation, particularly for independent nodes. The proposed layout and semantics disentanglement technique effectively capture both the spatial and interactive information of independent nodes. Taking the "lamp" and "stone" in the first image and the "shadow" in the second image as illustrations, these entities are neglected by previous methods, whereas our DisCo not only retains their semantic relevance but also infers their appropriate spatial placement within the scene. We also demonstrate the generalizable generation under consistency for graph manipulation (i.e., node addition and attribute control) in SG2I tasks, as shown in Figure 6.

**Ablation Study**. Table 4 explores the overall architecture by evaluating the alignment of the generated image with the objects and relationships depicted in the input scene graph. Following SGDiff [16], we conduct this analysis by graph-to-image (G2I) and image-to-graph (I2G) retrieval experiments.

Table 6: **Ablation study** for Multi-Layer Sampler (MLS).

| Method | IS ↑ | FID ↓ |
|---|---|---|
| Baseline (w/o MLS) | 20.5 | 23.0 |
| w/ LSD [25] | 21.1 | 22.7 |
| w/ MLS (Ours) | **22.3** | **21.9** |

Note that "w/o $D_s$" means processing each node embedding by MLP independently, instead of obtaining object interactive semantics through $D_s$. We observe that the spatial layout and interactive semantics collaborate to boost both retrieval tasks. These results demonstrate the effectiveness of integrating explicit spatial relations with implicit interactive semantics. In Table 5, we study the impact of different attention mechanisms. We inject graph conditions using different mechanisms: (a) Vanilla attention mechanism in the T2I diffusion model without our CMA; (b) CMA without attention mask **M**; (c) CMA with a union MLP after concatenating object and attribute embeddings; and (d) CMA with two separate MLPs before concatenating object and attribute embeddings. We found that CMA, which fuses separate encoding of object and attribute embeddings, significantly enhances the overall generation performance. Table 6 presents the Multi-Layer Sampler (MLS) ablation results, confirming its enhancement over the baseline and LSD [29]. In contrast to LSD, which randomly scrambles layouts, the proposed MLS naturally leverages a variety of coherent layouts and semantics produced by SL-VAE. Moreover, the increase in $ACC_{attr}$ scores also indicates that MLS facilitates controllability, especially in attribute control, while ensuring generation quality.

## 5 Related Works

**Diffusion Models**. Diffusion models (DMs) [2, 20, 31, 42] have achieved great success in high-quality image generation. The essence of DMs lies in estimating image distributions by iterative denoising noise-corrupted image, showcasing the superiority over VAEs [43, 44] and GANs [45] in training stability and likelihood estimation. To further explore the controllability of DMs, considerable efforts

have been devoted to conditional generation based on DMs. Benefiting from the naturalness of language [6] and the advancements of vision-language foundation models [6, 7, 8, 9], numerous text-to-image DMs [40, 41, 46] are beginning to emerge, facilitating explicit control of the corresponding semantics and style. However, the expressive capacity of linear text is limited. Therefore, many studies also endeavor to bolster global control through supplementary conditions, such as depth [46, 47], layout [10, 11, 12, 13], segmentation map [46, 48], and scene graph [14, 15].

**Image Generation from Scene Graphs**. Scene graphs are structured scene representations, where nodes represent objects and edges represent relationships between objects [17, 27]. Given the superiority of scene graphs over linear text in delineating multiple objects and their intricate relationships [27, 49, 50], many studies investigate image generation from scene graphs. These approaches typically fall into two categories: layout-based and semantics-based methods. Layout-based methods [17, 18, 19, 28, 36, 51] initially map scene graphs to coarse scene layouts comprising multiple bounding boxes and further refine these layouts to images with a layout-to-image model (e.g., Layout-Diffusion [10]). While the layout depicts spatial relationships, it fails to capture abstract relationships within the scene, leading to a lack of object interaction. Another branch is semantics-based methods [14, 16, 29, 52, 53], which focus on graph understanding by directly encoding semantic information from the scene graph. Nevertheless, these methods have limitations in addressing independent scene nodes, leading to issues like entity loss and unreasonable placement. In this paper, we propose a compositional image generation that leverages the layout and semantics derived from the scene graph representation. We complement explicit layout and implicit semantics to enhance the understanding of the diffusion model for scene graphs. Additionally, to improve the controllability in the scene-graph-to-image task, we also attain generalizable generation for object-level graph manipulation (i.e., node addition and attribute control).

# 6 Limitations

The proposed CMA injects object-level information into the diffusion model via masks from the layout, effectively mitigating semantic ambiguity and limiting attribute leakage. In scenarios involving object overlap, the proposed CMA inhibits direct interaction between the visual token and the object embedding along with its attributes. Nonetheless, the attribute information from the visual token inadvertently leaks into the overlapping region in subsequent layers. Hence, there may be attribute leakage among the objects, as shown in Figure 7.

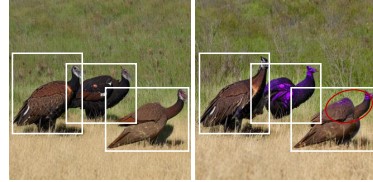

Figure 7: **Qualitative limitations** on attribute leakage of overlapping.

# 7 Conclusion

In this study, we leverage the disentangled textual scene graph representation to condition the diffusion process for generating complex scene images. The innovation of our framework lies in utilizing the VAE for scene relationship modeling and diffusion model (DM) for the composite visual generation. To comprehensively capture spatial relationships and non-spatial interactions within scenes, we introduce the Semantics-Layout Variational AutoEncoder (SL-VAE) for deriving diverse layouts and semantics from a single scene graph. Building upon them, we propose the Compositional Masked Attention (CMA) integrated with DM, which guides the de-noising trajectory by compositing extracted object-level graph information with fine-grained attributes. We also introduce a Multi-Layer Sampler (MLS) to preserve the main visual content while modifying the input scene graph. Extensive experiments demonstrate that our framework outperforms current methods conditioned by text, layout, or scene graph in relationship modeling and controllability.

# Acknowledgments

This research is supported by the National Natural Science Foundation of China [Grant 62302246] and the Zhejiang Provincial Natural Science Foundation of China [Grant LQ23F010008]. We also express our sincere gratitude to the AI Computing Center at the Eastern Institute of Technology for their valuable support and assistance.

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

# A   Appendix

This appendix is organized as follows: Section A.1 provides more details about Semantics-Layout Variation AutoEncoder; Section A.2 introduces the Stable Diffusion and its attention mechanism; Section A.3 describes the implementation of the Multi-Layer Sampler in detail; Section A.4 covers more ablation studies; Section A.5 presents more qualitative results, including comparison visualization and graph manipulation; Section A.6 delves into the broader societal impacts of this work. The core script is zipped and attached to the supplementary material.

## A.1   Semantics-Layout Variation AutoEncoder

Recall that we apply the triplet-GCN-based CVAE architecture in Section 3.1. Each triplet-GCN layer in the encoder and decoder takes the node and edge embeddings. Specifically, the $GCN_l$ mentioned in the paper uses two cascading MLPs $\{\mathrm{mlp}_1, \mathrm{mlp}_2\}$ to deal with node and edge embeddings:

$$(\psi_i^l, \phi_{ij}^{l+1}, \psi_j^l) = \mathrm{mlp}_1(\phi_i^l, \phi_{ij}^l, \phi_j^l), l \in \{0, \ldots, L-1\}, \tag{12}$$

$$\phi_i^{l+1} = \psi_i^l + \mathrm{mlp}_2(\mathrm{avg}(\psi_j^l \mid j \in \mathcal{N}_\mathcal{E}(o_i))), \tag{13}$$

where $l$ denotes the layer index of encoder or decoder, $\mathcal{N}_\mathcal{E}$ denotes the neighbor index set for each node, $\mathrm{avg}$ denotes the average pooling operation, $\phi$ and $\psi$ denote intermediate features. Hence, $\mathrm{mlp}_1$ conducts message passing among interconnected nodes and updates the edge features, while $\mathrm{mlp}_2$ aggregates features from all neighboring nodes and updates its features. For graph union encoder, we let $(\phi_i^0, \phi_{ij}^0, \phi_j^0) = (\mathcal{O}_i, \mathcal{E}_{ij}, \mathcal{O}_j)$. The last embedding $\phi_i^L$ is parameterized to the Gaussian distribution $Z \sim \mathcal{N}(\mu, \sigma)$, where $\mu, \sigma \in \mathbb{R}^{D_z}$ output by two additional MLPs and $D_z$ denotes the dimensional of latent space for node embedding.

## A.2   Diffusion with Compositional Masked Attention

**Stable Diffusion** [20] is one of most popular text-to-image model. As described in Section 2.1, Stable Diffusion uses a U-Net $\epsilon_\theta$ composed of convolution and transformer to estimate noise. The transformer includes two attention mechanisms, namely Cross-Attention, and Self-Attention.

**Cross-Attention Layer**. Text prompts are mapped to sequence embeddings by CLIP text encoder and integrated into UNet via Cross-Attention to guide the de-noising trajectory:

$$Attention(Q_{visual}, K_{text}, V_{text}) = softmax(\frac{Q_{visual}K_{text}^T}{\sqrt{d}}) \cdot V_{text} \tag{14}$$

where $Q_{visual}$ denotes the Query from the visual token of the UNet, $K_{text}$ and $V_{text}$ denotes Key and Value from text embeddings, all of which are projected by linear layers, $d$ denotes the dimension of $Q_{visual}$, $K_{text}$, and $V_{text}$.

**Self-Attention Layer**. Self-Attention captures self-related information within visual tokens:

$$Attention(Q_{visual}, K_{visual}, V_{visual}) = softmax(\frac{Q_{visual}K_{visual}^T}{\sqrt{d}}) \cdot V_{visual} \tag{15}$$

where $Q_{visual}$, $K_{visual}$, and $V_{visual}$ separately represent the Query, Key, and Value in self-attention layers, which are projected by linear layers. The self-attention mechanism isolates the information flow between specific tokens by multiplying a mask $\mathbf{M}$ to the $Q_{visual}K_{visual}^T$. Since $\mathbf{M}$ is applied before $softmax$, the value of the isolated position is set to negative infinity $-inf$.

**Compositional Masked Attention Layer**. Based on the attention mask $\mathbf{M}$ that depends on layout $\mathcal{B}$, the Compositional Masked Attention can be expressed as:

$$Attention(Q_{CMA}, K_{CMA}, V_{CMA}) = softmax(\frac{Q_{CMA}K_{CMA}^T \odot \mathbf{M}}{\sqrt{d}}) \cdot V_{CMA} \tag{16}$$

where $Q_{CMA}$, $K_{CMA}$, and $V_{CMA}$ individually represent the Query, Key, and Value derived from $\mathcal{V} \otimes \hat{\mathcal{C}}$, achieved through linear layer projections. We insert our proposed Compositional Masked Attention (CMA) between self-attention and cross-attention layers.

## A.3   Multi-Layer Sampler

**Layered Scene Representation**. We decompose a controllable scene containing $N_o$ objects into $N_o$ layers. Different from SceneDiffusion [25], our approach involves each layer incorporating not only

separate latent code $z_i$ and spatial layout $b_i$, but also integrating the interactive semantics $s_i$ produced by the SL-VAE. Here we convert the layout parameter $b_i$ to two parts: (1) a fixed *object-centric* binary mask $m_i \in \{0,1\}^{c \times w \times h}$ to solely show the geometric property of the object, and (2) a two-element offset $p_i = \{\mu_i, \upsilon_i\}$ to solely indicate its spatial locations, with $\mu_i$ and $\upsilon_i$ defining the horizontal and vertical movement range. We sample Gaussian noise individually for the initial latent code of each layer, i.e., $\mathcal{Z} = \{z_i^{(T)} \sim \mathcal{N}(0,1)\}_{i=1}^{N_o}$. Then we utilize the layout-converted non-overlapping masks $\{l_i\}_{i=1}^{N_o}$ to derive the aggregated latent code $z$ from various layers:

$$z^{(t)} = \sum_{i=1}^{N_o} l_i \odot \overline{shift}(z_i^{(t)}, p_i) \tag{17}$$

$$l_i = \overline{shift}(m_i, p_i) \prod_{j=1}^{N_i-1} (1 - \overline{shift}(m_j, p_j)), \tag{18}$$

where $\odot$ denotes element-wise multiplication, and $\overline{shift}(x, p)$ denotes spatially shifting the values of $x$ in the direction of $p$.

**Multi-Layer Generation**. We introduce the Multi-Layer Sampler that matches our diverse layout and semantic simulation. In contrast to SceneDiffusion [25] which scrambles the reference layouts randomly, we sample additional $N_l$ layouts and semantics by the proposed SL-VAE. On the one hand, the SL-VAE ensures that the generated scene layout is reasonable. On the other hand, we take full advantage of the paired object-level *(layouts, semantics)*. Specifically, the denoising scheme consists of four steps:

**(a)** Sampling additional $N_l$ layouts $\{\mathcal{B}_n = \{b_{n,i}\}_{i=1}^{N_o}\}_{n=1}^{N_l}$ and semantics $\{\mathcal{S}_n = \{s_{n,i}\}_{i=1}^{N_o}\}_{n=1}^{N_l}$ by the proposed SL-VAE. Note that $N_l$ fixed seeds exist for the same scene graph. According to the description of the layered representation, we convert the layout to get offset $\{\mathcal{P}_n = \{p_{n,i}\}_{i=1}^{N_o}\}_{n=1}^{N_l}$.

**(b)** Aggregating latent codes from various layers in each scene:

$$z_n^{(t)} = \sum_{i=1}^{N_o} l_i \odot \overline{shift}(z_i^{(t)}, p_{n,i}) \tag{19}$$

**(c)** Estimating the noise $\hat{\epsilon}_n^{(t)}$ from each aggregated latent code $z_n^{(t)}$ and gets denoised aggregated latent code $\hat{z}_n^{(t-1)} \in \{\hat{z}_1^{(t-1)}, \dots, \hat{z}_{N_l}^{(t-1)}\}$:

$$\hat{\epsilon}_n^{(t)} = \sum_{i=1}^{N_o} m_{n,i} \odot \epsilon_\theta(z_n^{(t)}, E_{\text{CLIP}}(o_i), b_{n,i}, s_{n,i}, a_i, t), \tag{20}$$

where $m_{n,i}$ is the non-overlapping mask converted by the layout $b_{n,i}$.

**(d)** Updating the latent code of each layer by computing the weighted average of the $N_l$ aggregated latent code

$$z_i^{(t-1)} = \frac{\sum_{n=1}^{N_l} \overline{shift}(l_i \odot \hat{z}_n^{(t-1)}, -p_{n,i})}{\sum_{n=1}^{N_l} \overline{shift}(l_i, -p_{n,i})} \tag{21}$$

where $\overline{shift}(x, -p)$ denotes spatially shifting the values of $x$ in the reverse direction of $p$.

### A.4 More Ablation Studies

**Graph Construction**. We conduct ablation for graph construction in Table 7. We investigate the impact of different graph components (i.e., CLIP, Box, and Learnable Embeddings) by turning off each independently. We observe that each component improves the performance, all of which are crucial components presented in our DisCo.

Table 7: **Ablation study** for graph construction.

| Graph Type | IS $\uparrow$ | FID $\downarrow$ |
|---|---|---|
| No CLIP Emb. | 20.6 | 23.9 |
| No Box Emb. | 21.7 | 22.5 |
| No Learnable Emb. | 21.9 | 22.2 |

**Computing Consumption.** We demonstrate the impact of our proposed CMA on the computational complexity of the U-Net within the Stable Diffusion, as presented in Table 8. We use Floating Point

Table 8: **Ablation study** for computing consumption.

| Method | FLOPs (G) | Params (M) | Time (ms) |
|--------|-----------|------------|-----------|
| SD-v1.5 [20] | 677.5 | 859.4 | 37.9 |
| DisCo | 724.1 | 875.8 | 108.3 |

Operations (FLOPs), the number of parameters (Params), and inference time (Time) to measure computing consumption. The FLOPS and Time metrics are conducted by processing the tensor with a resolution of $2 \times 4 \times 64 \times 64$ on an NVIDIA A100 GPU. Our proposed DisCo significantly improves the controllability of the Stable Diffusion with a tolerable increase in computational cost.

### A.5 More Visualization Results

Figure 8 showcases more generalizable generation results under consistency for graph manipulation (i.e., node addition and attribute control) in SG2I task. In Figure 9, 10, and 11, we present more visualization comparisons with the methods conditioned by text, layout, or scene graph, which demonstrates the superiority of our DisCo in terms of generation rationality and controllability.

### A.6 Broader Impacts

We demonstrate the superiority of our DisCo over existing generation methods based on text, layout, and scene graphs, suggesting a potential beneficial influence on the realms of art creation and data synthesis. Nevertheless, there remains a concern regarding the possibility of generating malicious images or infringing copyright.

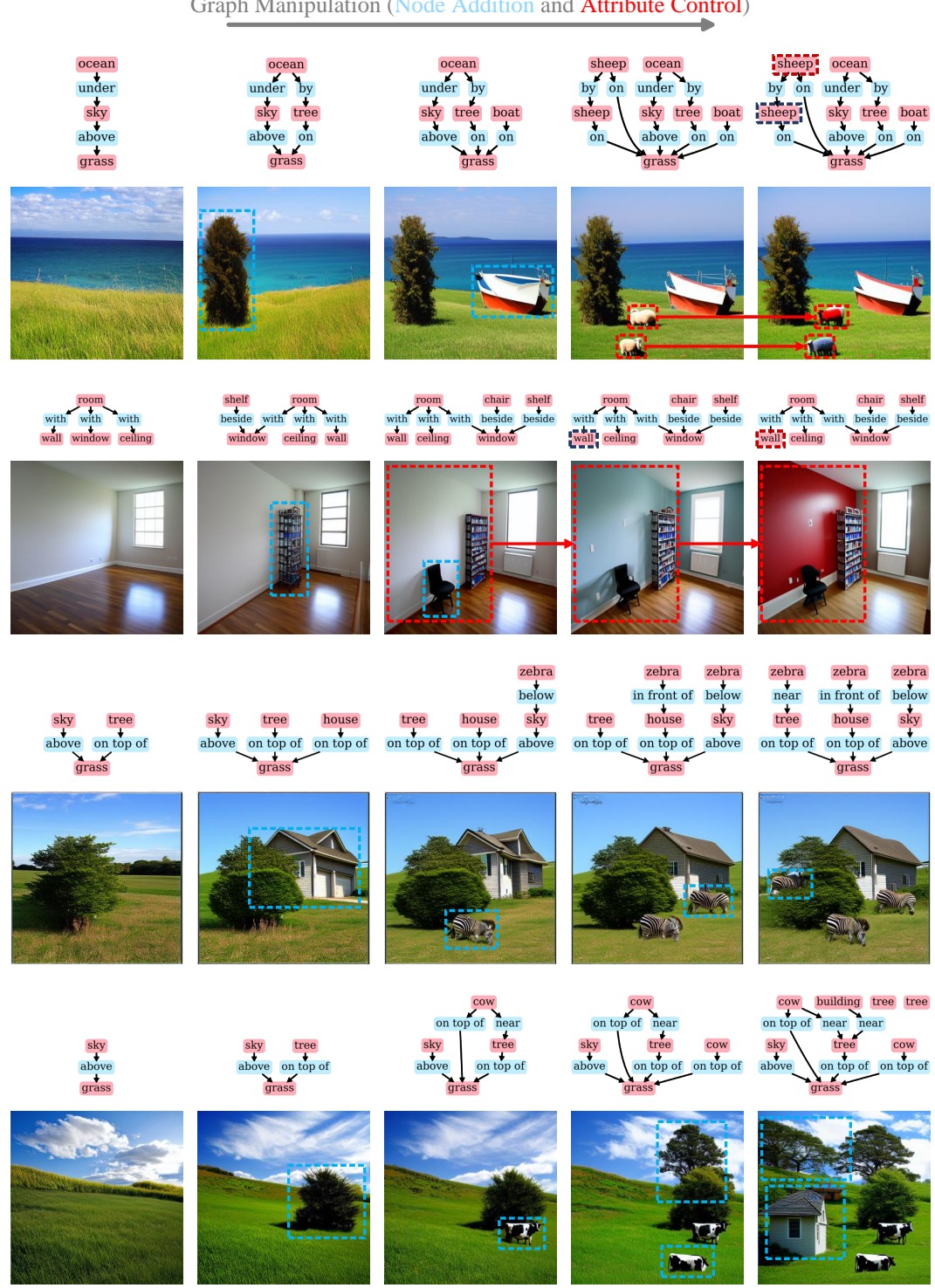

Figure 8: **Generalizable Generation Samples** under Consistency for Graph Manipulation.

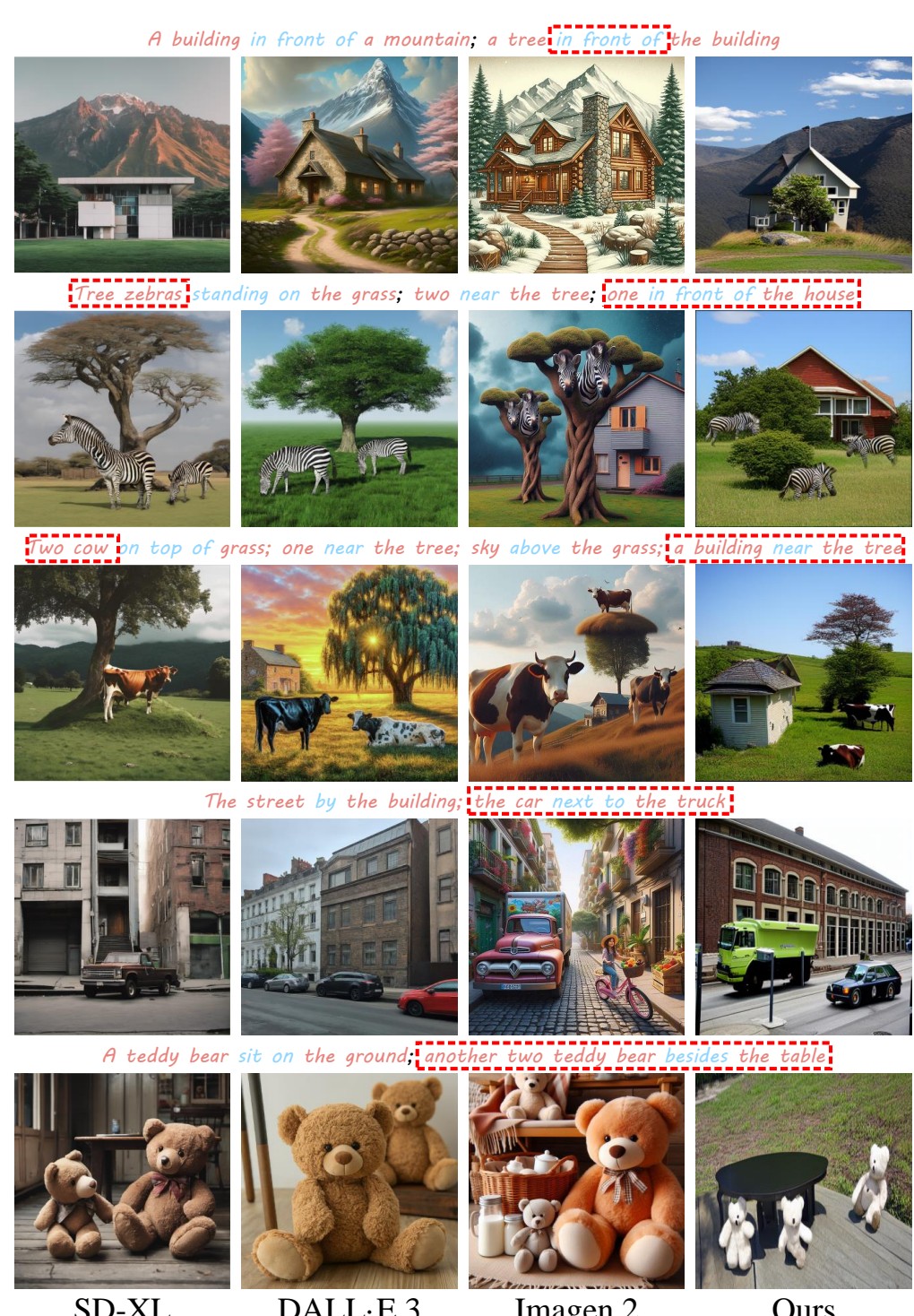

Figure 9: **Qualitative Comparison** with Text-to-Image methods.

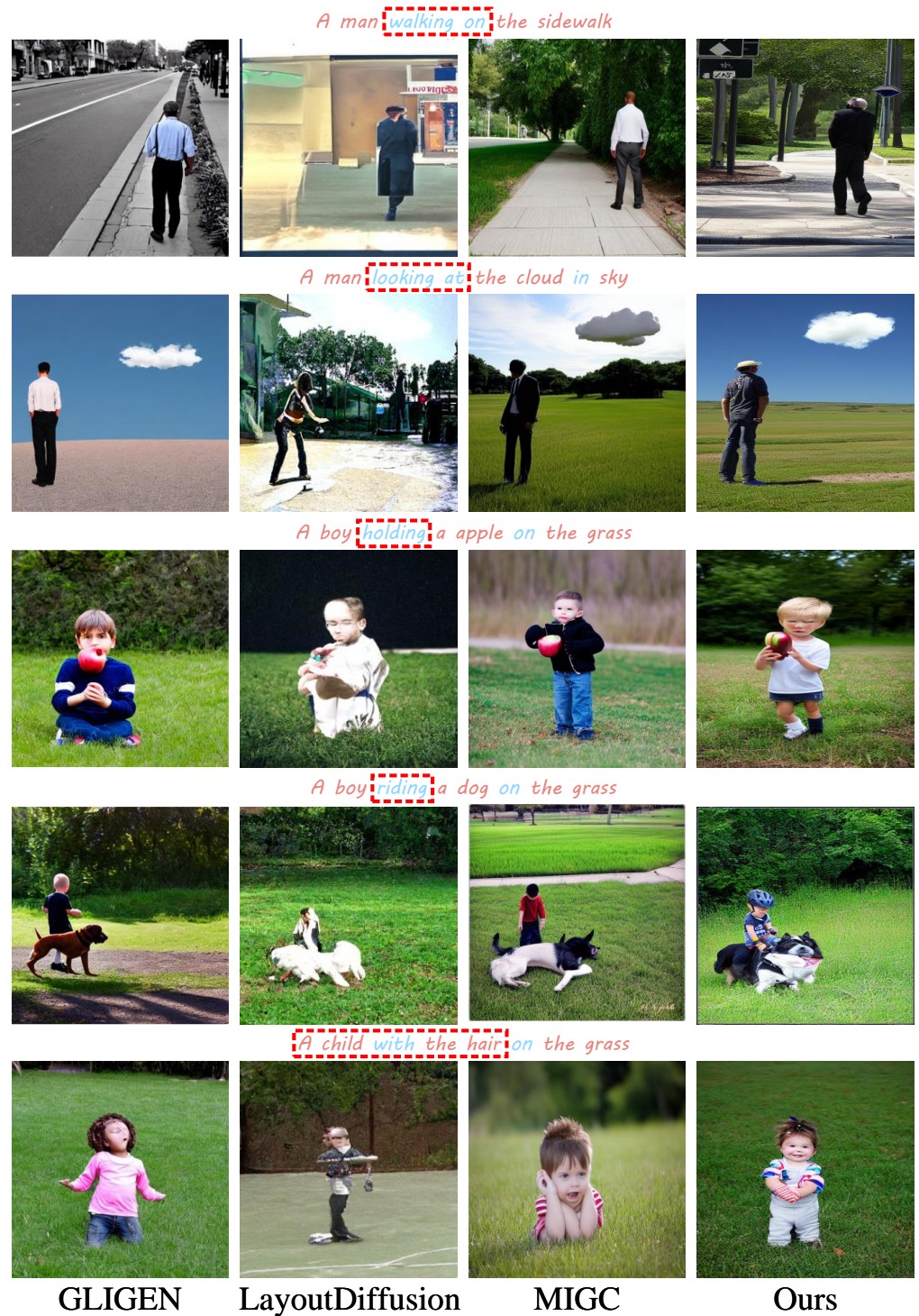

Figure 10: **Qualitative Comparison** with Layout-to-Image methods.

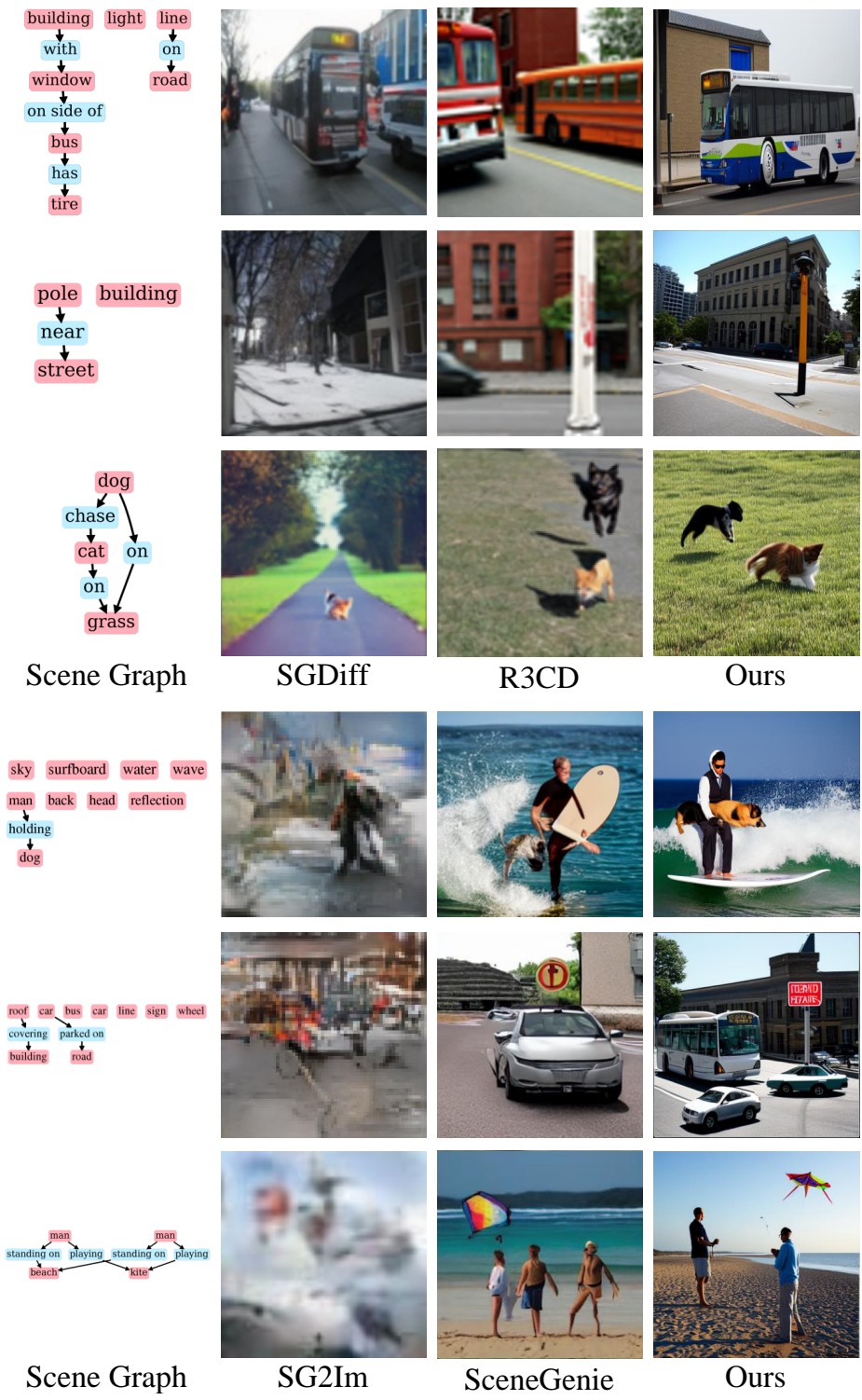

Figure 11: **Qualitative Comparison** with Scene-Graph-to-Image methods.

