# OpenReview forum: "Scene Graph Disentanglement and Composition for Generalizable Complex Image Generation"
_NeurIPS.cc/2024/Conference — NeurIPS 2024 spotlight_

### Official Review · Reviewer_Knts · 2024-07-08

**Soundness:** 4
**Presentation:** 3
**Contribution:** 3
**Rating:** 8
**Confidence:** 4

**Summary:**

This paper employs scene graph for image generation. Different from the previous methods, they employ the generative capabilities of variational autoencoders and diffusion models in a generalizable manner, compositing diverse disentangled visual clues from scene graphs. The authors propose a semantics-Layout Variational AutoEncoder to jointly derive layouts and semantics from scene graph. Then they develop CMA integrated with a diffusion model. They also introduce the multi-layered sampler for achieving graph manipulation. Experiments show that the method outperforms existing methods.

**Strengths:**

1. The paper address the problems in existing methods well. Existing methods in the field of scene graph to image generation mainly depends on the layout or semantics. Using one of them may cause some problems. Inspired by these phenomenons, the authors propose the method to jointly considering the layout and the semantics. What's more, the techniques used in the framework are novel enough.
2. The authors conduct plenty of experiments. The ablation studies support the motivations.

**Weaknesses:**

The only weakness I found is that the authors should reorganize the paper carefully. The writings is not so clear in some sections. For example, the multi-layered sampler section is too abstract to be understood.

**Questions:**

No.

**Limitations:**

Yes.

---

> ### Author Rebuttal · Authors · 2024-08-04
>
> Thank you for your positive comments and valuable feedback on our work! We are excited and encouraged by your support! Below we address your concern.
>
> **Q1: About the clarification of the multi-layered sampler section.**
>
> **R1**: We sincerely appreciate the valuable suggestion, and we will make our method more understandable in the revision. For the multi-layered sampler, we define each object as a layer, thus allowing independent object-level Gaussian sampling. Besides, we leverage diverse scene conditions ($N_l=5$) disentangled by SL-VAE for locally conditioned diffusion and aggregate them into the latent code for each object. In summary, it can be simply understood as calculating a more **robust layered representation** for each object in the scene, which achieves an “isolated” image editing effect.

---

> > ### Comment · Reviewer_Knts · 2024-08-12
> >
> > After reading the responses and comments from other reviewers, most concerns have been addressed. I am especially interested   of the anti-logical relationships generation. I keep my rating considering that this paper deserves being accepted. I also hope the authors could release the codes as soon as possible.

---

### Official Review · Reviewer_SBq5 · 2024-07-12

**Soundness:** 3
**Presentation:** 3
**Contribution:** 3
**Rating:** 6
**Confidence:** 4

**Summary:**

The paper proposes DisCo (Disentangled Compositional image generation), which integrates both layout and semantic information derived from scene graphs to improve the quality and controllability of generated images. In particular, DisCo has three main components: Semantics-Layout Variational AutoEncoder (SL-VAE) for disentangling spatial layouts and semantics, Compositional Masked Attention (CMA) for fine-grained attribute guidance, and Multi-Layered Sampler (MLS) for object-level graph manipulation. Extensive experiments demonstrate that DisCo outperforms state-of-the-art methods in generating rational and controllable images from text, layout, and scene graph conditions.

**Strengths:**

1. The motivation is clear. The idea of disentangling layout and semantics from scene graphs is novel.
2. DisCo outperforms recent methods in both fidelity and diversity of image generation, as evidenced by the IS and FID scores. Overall, it enhances the generation diversity and controllability.
3. Extensive experiments and ablation studies have demonstrated the effectiveness and the contribution of each component.

**Weaknesses:**

1. The increased inference cost of DisCo (Table 7). In particular, CMA mechanism might increase the computational cost, which may limit the method's scalability and efficiency, especially for large-scale applications. Moreover, since diffusion models are already quite large, the additional AutoEncoders (Lines 129-130) may result in more parameter and memory overhead.
2. DisCo requires expensive training, e.g. 4 A100 GPUs, as indicated in Lines 202-203. With more models releasing recently, this technique might be not scalable.
3. The image quality looks better with this proposed method. However, as metrics today cannot always reflect the real image quality, it would be more convincing to conduct a user study, e.g. votes, to quantify the advantage of DisCo compared to previous works.

**Questions:**

See the weakness.

**Limitations:**

Yes, authors have stated their limitations on attribute leakage of overlapping in Section A.6, as well as a short discussion on the broader impact in Section A.7.

Beside that, authors are also encouraged to add a few discussions on the efficiency of their proposed pipeline. Even though these overheads are inevitable (they are quite common in most researches), a clearer trade-off between the improved image quality and the increased model complexity would help to better assess the value of this work.

---

> ### Author Rebuttal · Authors · 2024-08-04
>
> We greatly appreciate all of your valuable suggestions, which play a pivotal role in enhancing the quality of our paper. Bellow we address all your concerns.
>
> **Q1: Discussion about the complexity and quality.**
>
> **R1**: Thanks for your valuable suggestions. To comprehensively evaluate the complexity and efficiency of our DisCo, we here provide a clearer trade-off between the improved image quality and the increased model complexity to help readers better assess the value of our work. We evaluate the image quality using the T2I-CompBench [35] and measure model complexity using Floating Point Operations (FLOPs) and the number of parameters (Params). The results are shown in the following table.
>
> |||||||
> |--------------|:----:|:----:|:----:|:----:|:----:|
> |**Method**|**UniDet**|**B-VQA**|**3-in-1**|**FLOPs (G)**|**Params (M)**|
> |SD-v1.5|0.1246|0.3765|0.3080|677.5|859.4|37.9|0.3080|677.5|859.4|
> |**DisCo**|**0.2376**|**0.6959**|**0.4143**|732.9|894.6|
> |**(ours)**|**(+85.9%)**|**(+84.8%)**|**(+34.5%)**|(+8.2%)|(+4.1%)|
>
> As we can see that our proposed DisCo achieves significant image quality improvements with a tolerable increase in computational cost.
>
> Besides, for the additional AutoEncoders (Lines 129-130) you mentioned, we rectify that this design would not bring more parameter and memory overhead, just negligible increase:
>
> ||||
> |--------------|:----:|:----:|
> |**Method**|**FLOPs (G)**|**Params (M)**|
> |**DisCo (w/o SL-VAE)**|724.1|875.8|
> |**DisCo (w/ SL-VAE)**|732.9 **(+1.2%)**|894.6 **(+2.1%)**|
>
>
> **Q2: About training cost and scalability.**
>
> **R2**: We present the training cost between different methods in the following table. We use the A100-80G GPU hours as the metric.
>
> ||||||
> |:----:|:----:|:----:|:----:|:----:|
> |**Method**|GLIGEN|MIGC|R3CD|**DisCo (ours)**|
> |**GPU hours**|~120|~300|~180|**~90**|
>
> It can be found that that our proposed DisCo has lower training cost compared to other methods. Actually, our DisCo can be treated as a plug-in controller for other models, once we release the weights of SL-VAE and CMA upon paper acceptance, users could directly fine-tune models on GPUs like **3090-24G** (no need training from scratch) to make our DisCo scalable to more scenarios.
>
> **Q3: The metrics to reflect the real image quality, and the required user study.**
>
> **R3**: We sincerely appreciate the valuable suggestion. Actually, we evaluate our method on T2I-CompBench, where the metrics (i.e., the spatial/non-spatial relationships, attributes, and complex scenes) are validated to be consistent with human assessments [35].
>
> However, we firmly believe that the user study you mentioned is more convincing. Therefore, we conduct a user study by recruiting 50 participants from Amazon Mechanical Turk. We randomly select 8 prompts for each method, resulting in 80 generated images. We ask participants to score each generated image independently based on the image-prompt alignment. The worker can choose a score from {1, 2, 3, 4, 5} and we normalize the scores by dividing them by 5. We then compute the average score across all images and all workers. The results of user study are presented in the table below.
>
> ||||||||||||
> |:----:|:----:|:----:|:----:|:----:|:----:|:----:|:----:|:----:|:----:|:----:|
> |**Method**|SD-XL|DALL$\cdot$E 3|Imagen 2|GLIGEN|LayoutDiffusion|MIGC|SG2Im|SGDiff|R3CD|**DisCo (ours)**|
> |**Alignment Score**|0.6684|0.5944|0.5637|0.6549|0.6200|0.7055|0.3783|0.4717|0.6928|**0.8533**|

---

> > ### Comment · Reviewer_SBq5 · 2024-08-12
> > **Official Comment from Authors**
> >
> > Thanks for the authors' rebuttal. They have addressed my concerns. Thus, I would like to raise my score to 6.

---

### Official Review · Reviewer_anRy · 2024-07-14

**Soundness:** 3
**Presentation:** 3
**Contribution:** 3
**Rating:** 6
**Confidence:** 4

**Summary:**

This paper presents "DisCo," a novel framework for generating complex images from structured scene graphs. Unlike traditional text-to-image or layout-to-image methods, DisCo utilizes a Semantics-Layout Variational AutoEncoder (SL-VAE) to disentangle and generate diverse spatial layouts and interactive semantics from scene graphs. It incorporates these elements using a Compositional Masked Attention (CMA) mechanism within a diffusion model, enhancing generation control and rationality. The framework also introduces a Multi-Layered Sampler (MLS) for flexible, graph-based image editing, preserving visual consistency while manipulating object attributes and positions.

**Strengths:**

1. Introduces innovative methods for disentangling and integrating spatial and semantic information from scene graphs, which is a novel approach in image generation.
2. Offers significant improvements in image generation from complex scene graphs, enhancing both the fidelity and controllability of generated image

**Weaknesses:**

1. The paper lacks quantitative comparisons with closely related baselines, such as R3CD, which could provide a more comprehensive evaluation of the model's performance. Inclusion of these comparisons could help validate the proposed advantages of DisCo over existing methods, particularly in handling complex scene graph-to-image generation tasks.
2. Some generated images, particularly those highlighted in Figure 10, exhibit unnatural aspect ratios and stretched elements, suggesting issues with the model’s handling of object proportions and spatial embeddings.
3. It would be great to discuss the scalability aspects, particularly how the proposed model handles graph sizes that exceed typical training configurations.
4. how the model performs with imperfect or noisy scene graphs, which are common in automatically extracted data.

**Questions:**

1. The paper presents results on standard benchmarks. However, can you provide insights or preliminary results on how the model performs across datasets with higher variability in object complexity and scene density?
2. Why were certain closely related baselines omitted from quantitative comparisons? Could inclusion of these baselines provide a more comprehensive evaluation?

**Limitations:**

The authors address the challenge of attribute leakage in overlapping objects, which affects image fidelity in scenes with dense object interactions. While mitigation strategies are discussed via the CMA mechanism, further refinement is required to eliminate this issue completely. Additional exploration into the computational efficiency and scalability of the proposed methods would also benefit the paper, providing a more comprehensive view of their practical applications and limitations.

---

> ### Author Rebuttal · Authors · 2024-08-04
>
> We sincerely appreciate the affirmation from the reviewer for our work. It serves as a strong motivation for us! Bellow we address your concerns sequentially.
>
> **Q1: More quantitative comparisons with related baselines, such as R3CD.**
>
> **R1**: Actually, in Table 1 of the manuscript, we have already provided the quantitative comparisons with the related SG2I baselines, including diffusion-based (e.g., R3CD) and GAN-based methods (e.g., SG2Im). We can see that our DisCo achieves the best in both IS and FID scores.
>
> Besides, we conduct a user study by recruiting 50 participants from Amazon Mechanical Turk.  We randomly select 8 prompts for each method, and ask participants to score each generated image independently based on the image-prompt alignment. The worker can choose a score from {1, 2, 3, 4, 5} and we normalize the scores by dividing them by 5. We then compute the average score across all images and all workers. The results of user study are presented in the table below.
> |||||||
> |:----:|:----:|:----:|:----:|:----:|:----:|
> |**Method**|SG2Im|SGDiff|SceneGenie|R3CD|**DisCo (ours)**|
> |**Alignment Score**|0.3783|0.4717|0.5032|0.6928|**0.8533**|
>
> **Q2: Unnatural aspect ratios issue in Figure 10.**
>
> **R2**:  Thank you for the questions raised by the reviewer. Actually, this is due to typographical error of the manuscript, and we will fix it in revison.
>
> Nevertheless, we can still easily see from Figure 10 that our DisCo has a superior ability in modeling **non-spatial relationships**, such as human "holding" and "looking at" in Figure 10.
>
> **Q3: Scalability aspect of graph size, object complexity and scene density.**
>
> **R3**: Great point! We sincerely appreciate the valuable suggestion, and we will discuss the scalability aspects in the revision. Actually, during the training of our SL-VAE, the way we used that forms **a batch of graphs with different node quantities** is to merge them into one larger graph, where each subgraph is not connected to each other (similar to independent nodes). Therefore, the GCN-based encoder can effectively learn relationships independent of the number of nodes, and thus handle graph sizes that exceed typical training configurations.
>
> Besides, we also provide the generation results with higher variability in object complexity and scene density. Please refer to **Figure A1 of the newly added PDF file** for more information, which demonstrates that our proposed model could handle larger graph sizes well.
>
> **Q4: How the model performs with imperfect or noisy scene graphs?**
>
> **R4**: Thank you for the questions raised by the reviewer. Actually, our method performs robust enough and is not sensitive to the imperfect or noisy scene graphs. For example, we consider two cases of imperfect or noise scene graphs, i.e., the anti-logical and the missing relationships.
>
> - **Anti-logical relationship**. There may occur anti-logical relationship in the provided scene graph, such as “*a dog on the sky*” and “*a sheep on top of a tree*”. As stated in the manuscript, our DisCo disentangles spatial relationships and interactive semantics. The layouts that represent the spatial relationships can output relative positions that follow the description even with an anti-logical input. Besides, the embeddings that represent the interactive semantics influence visual semantics with the given anti-logical relationship while ensuring semantic correctness. Thus, our method could still generate reasonable outputs matching the descriptions even if they are anti-logical. We present the anti-logical generation sample in **Figure A2 the newly added PDF file**.
>
> - **Missing relationships**. In case of node relationship missing (as we discussed in Figure 6 (c) of the manuscript), we deliberately define a special node "__#image#__" and a special relationship "__#in#__", which represent the whole scene and the relationship of objects to the scene, respectively. All regular nodes are connected to the "__#image#__" node with the "__#in#__" edge. Thus, independent nodes (i.e., missing relationship) could still interact with other nodes in the same scene, and infer their placement based on mutual semantics. In addition, our explicit spatial layout further ensures that there are no node missing problems. We present the results of the missing relationships in **Figure A3 of the newly added PDF file**. We will add these details in revision.

---

> ### Author Response · Authors · 2024-08-13
>
> Thank you for your continued efforts. We have provided comprehensive rebuttals and tried to address the concerns raised in your reviews. Please take the time to review if possible, if you have any further questions or require additional clarification, please let us know and we welcome discussions in any format. Thanks again.

---

### Official Review · Reviewer_3rPJ · 2024-07-17

**Soundness:** 3
**Presentation:** 3
**Contribution:** 2
**Rating:** 5
**Confidence:** 3

**Summary:**

This paper proposes a method that uses a scene graph and integrates variational autoencoders (VAEs) and diffusion models to address complex scene generation. Specifically, a Semantics-Layout Variational AutoEncoder (SL-VAE) is used to derive diverse layouts and semantics from the scene graph, while a Compositional Masked Attention (CMA) combined with a diffusion model incorporates these elements as guidance. Additionally, a Multi-Layered Sampler (MLS) is introduced for isolated image editing. Experiments show that this approach outperforms recent methods in generation rationality and controllability.

**Strengths:**

1. This paper considers an important issue in text-to-image generation realm.
2. The structure design in Section 3 makes sense.
3. The experimental results shown in Table 1 and 2 show the effectiveness of this method.

**Weaknesses:**

1. My main concern is the practical application of this method. As we all known, scene graph building is not a trivial task, but you don't explain detail in the paper how to construct an exact scene graph. In addition, during inference, the prompt proposed by uses may be non-standard so that building a scene graph may be more difficult.
2. Besides, recent SOTA models, e.g. DALLE3, stable diffusion 3 try to solve the complex generation task by large-scale fine-grained dataset construction. How do you compare your methods with these data-centric methods. The authors should spend more space discussing the issues.

**Questions:**

Please see the weakness

**Limitations:**

Please see the weakness

---

> ### Author Rebuttal · Authors · 2024-08-04
>
> We greatly appreciate all of your valuable suggestions, which play a pivotal role in enhancing the quality of our paper. Below we address all your concerns.
>
> **Q1: Details of scene graph construction.**
>
> **R1**: Thank you for the questions raised by the reviewer, and we will add the details of scene graph construction from text in the revision. Actually, our DisCo focuses on scene-graph-to-image (SG2I) task, which typically assumes that the scene graph is ready [14, 17]. However, to make our method more practical and allow comparison with the popular text-to-image (T2I) methods, we also provide a text-to-scene graph conversion method (it is not the focus of our paper, so we didn't add it in the original manuscript), which is feasible and convenient with the assistance of the recent powerful LLM such as GPT-4o or Llama.
>
> Specifically, as mentioned in Section 2.2, we use LLM to extract **instances (subject and object)** and **relationships (predicates)** from the prompt. Given a text describing the scene, the LLM outputs a structured format that contains **a node list** and **a triple list**. To standardize the output format of the LLM and facilitate inferring scene graph from non-standard prompt, we here provide a simplified template of our instructions and in-context examples for scene graph building:
>
> ```1. Task instruction``` **```(pre-provided)```**
>
>     You are a master of composition who excels at extracting key objects and their relationships from input text. You should reorganize the input text as a scene graph format, which consist of a node list representing the objects and a triple list representing the relationships between the objects.
>
> ```2. Scene graph construction tutorial``` **```(pre-provided)```**
>
>     There are a few rules to follow:
>         # Output format specification
>         # Numbering rules of the same category
>         # Extract relationships from non-standard text based on context
>
> ```3. In-context examples``` **```(pre-provided)```**
>
>     **Prompt Example**:
>         Two men standing on the beach, both of them playing a kite.
>
>     **Output Example**:
>         Objects: [man1, man2, beach, kite]
>         Relationships: [(man1, standing on, beach), (man2, standing on, beach), (man1, playing, kite), (man2, playing, kite)]
>
>     **More examples using non-standard prompt**
>
> ```4. Trigger reasoning ability of LLM``` **```(user-provided)```**
>
>     **User Prompt**
>         A sheep by another sheep on the grass with the ocean under the sky; the ocean by a tree; a boat on the grass
>
>     **Output**:
>         Objects: [sheep1, sheep2, grass, ocean, sky, tree, boat]
>         Relationships: [(sheep1, by, sheep2), (sheep1, on, grass), (sheep2, on, grass), (grass, with, ocean), (ocean, under, sky), (ocean, by, tree), (boat, on, grass)]
>
> Note that such scene graph built by LLM only applies to the inference phase for practicality. For training, we use specialized scene graph-to-image datasets (Visual Genome [27] and COCO-Stuff [26]) where the required information like box labeling is all well covered.
>
> **Q2: Comparison with recent SOTA data-centric text-to-image (T2I) methods.**
>
> **R2**: Actually, we have discussed and compared with these methods in Figures 1 (a) and 6 (a) of the manuscript. These data-centric text-to-image (T2I) methods like DALLE$\cdot$3 typically benefit from large-scale text-image pairs that can be automatically collected in high quality. However, even with large-scale fine-grained text annotation, they still suffer from deficiencies in aspects such as **quantity generation** and **relationship binding** due to the **linear structure** of the text.
>
> In summary, we point out the advantages of our **structured scene graphs** over these data-centric text-only methods in representing complex scenes from the following aspects:
>
> - Based on scene graph data, which is more compact and efficient than text, our DisCo demonstrates advantages in generation **rationality** and **controllability**, particularly for quantity generation and relationship binding. The qualitative  comparisons with the T2I models are shown in Figures 1 (a) and 6 (a) of the manuscript.
>
> - We also conduct evaluation with these T2I methods on T2I-CompBench [35], which quantifies our improvement in **spatial/non-spatial relationships**, **attributes**, and **complex scenes**. The results are shown in the table below (**bold** for 1st, *italic* for 2nd).
>
>     | **Method**|**UniDet**|**CLIP**|**B-VQA**|**3-in-1**|
>     |--------------|:----:|:----:|:----:|:----:|
>     | SD-v1.4|0.1246|0.3079|0.3765|0.3080|
>     | SD-v2| 0.1342 |0.3127|0.5065|0.3386|
>     | SD-XL| 0.2032 |0.3110|0.6369|0.4091|
>     | Pixart-$\alpha$|0.2082|*0.3179*|0.6886|*0.4117*|
>     | DALL$\cdot$E 2|0.1283|0.3043|0.5750|0.3696|
>     | DALL$\cdot$E 3|*0.2265*|0.3003|**0.7785**|0.3773|
>     | **DisCo (ours)**|**0.2376**|**0.3217** |*0.6959*|**0.4143**|
>
> - Besides, we conduct a user study by recruiting 50 participants from Amazon Mechanical Turk. We randomly select 8 prompts for each method, and ask participants to score each generated image independently based on the **image-prompt alignment**. The worker can choose a score from {1, 2, 3, 4, 5} and we normalize the scores by dividing them by 5. We then compute the average score across all images and all workers. The results of user study are presented in the table below, and we can see our DisCo achieves the best alignment score.
>     ||||||
>     |:----:|:----:|:----:|:----:|:----:|
>     |**Method**|SD-XL|DALL$\cdot$E 3|Imagen 2|**DisCo (ours)**|
>     |**Alignment Score**|0.6684|0.5944|0.5637|**0.8533**|
>
> - Moreover, the SL-VAE designed in our DisCo disentangles diverse conditions from the scene graph, which allows multi-round “**separate object-level manipulation while keeping the other content unchanged**” editing effect as a by-product that is more practical than the general T2I models. The generalizable generation results are illustrated in Figures 5 and 8 of the manuscript.

---

> > ### Comment · Reviewer_3rPJ · 2024-08-12
> > **Thanks for your kind explanation**
> >
> > After thoroughly reading your rebuttal and other reviewers' comments, I will raise my score to 5

---

### Author Rebuttal · Authors · 2024-08-04

Dear reviewers,

We thank all reviewers for their time and efforts in reviewing our paper. These constructive reviews can bring multiple improvements to our manuscript. We are encouraged that the reviewers appreciate our method, including:

 - structure design that makes sense *[Reviewer 3rPJ]*
 - innovative method *[Reviewer anRy, SBq5, Knts]*
 - clear motivation *[Reviewer SBq5, Knts]*
 - outperform prior methods in rationality and controllability *[Reviewer 3rPJ, anRy, SBq5]*

We have also made diligent efforts to address all the concerns raised point by point. Please see separate responses for details.

We are open to discussions and addressing any issues from reviewers. Your constructive comments can further help us to improve our method.

Sincerely yours,

Authors

---

### Decision · Program_Chairs · 2024-09-25

**Decision:**

Accept (spotlight)

**Comment:**

This paper received generally positive reviews during the initial review period. Some concerns were raised during that period, such as the potential overhead in application, comparison with data-centric models, and comparison with related baselines. The author's response effectively addressed many of these concerns. I agree with the reviewers and recommend acceptance. The authors are encouraged to include their responses from the rebuttal in the final version for completeness.